# Exploring Extension Implications for Slow Food Development in Iran: A Comprehensive Analysis

**Hassan Nazifi [1], Mohammad Sadegh Sabouri [1], Mohammad Sadegh Allahyari [2,*] , Mehrdad Niknami [1] and Elham Danaei [1]**

[1] Department of Agricultural Extension Education, Garmsar Branch, Islamic Azad University, Garmsar 3581631167, Iran; nazifi.hassan2@gmail.com (H.N.); m.sabouri@semkarshenas.ir (M.S.S.); m.niknami@iau-garmsar.ac.ir (M.N.); e.danaee@iau-garmsar.ac.ir (E.D.)

[2] Department of Agricultural Management, Rasht Branch, Islamic Azad University, Rasht 4147654919, Iran

\* Correspondence: allahyari@iaurasht.ac.ir

**Abstract:** This research aimed to ascertain the prerequisites for the advancement of the slow food movement in Iran. Employing both quantitative and qualitative methods, it adopted a descriptive and survey-oriented design. Semi-structured interviews were conducted with 15 experts well-versed in the extension of slow food, employing a snowball sampling technique. The interview data underwent coding and analysis employing open coding, axial coding, and selective coding methods. The study encompassed experts and managers in agricultural extension and education across the nation. For statistical analysis, a structural equation model and confirmatory factor analysis were employed, utilizing SMART PLS 3 and SPSS 26 software. The goodness-of-fit index (GoF) was utilized to evaluate the comprehensive validity of the research model. From a qualitative perspective, six primary facets of the slow food model emerged: 1. Extension strategies in harmony with slow food principles; 2. Methods of extending the slow food movement; 3. Supportive policies for slow food propagation; 4. Intervening conditions; 5. Causal conditions (triggers and applications) of the slow food paradigm; and 6. Outcomes resulting from the adoption of the slow food ethos. These facets collectively comprised a total of 38 sub-components. Through analysis of the structural equation model, key facets with substantial operational weight and significant influence on the promotion of slow food were identified. These prominent components encompass disease prevention, the organization of festivals and exhibitions, the revision of laws, the shaping of individuals' lifestyles, the enhancement of food tourism capacity, and the optimization of human resources.

**Keywords:** slow food; good; clean; fair; gastronomy; advisory system; sustainability

## 1. Introduction

Food plays a vital role in human existence, representing cultures and nations and fostering connections among individuals [1,2]. Across societies, varying food preferences, dining customs, and cultural traditions shape personal and collective identities [3]. The global nutritional transition reflects evolving patterns in food consumption [4]. The slow food movement's emergence is significant, advocating for quality, clean, and equitable food for a promising future, supporting sustainability and local agriculture [5,6].

Proponents of the slow food movement advocate a return to traditional, community-based, and organic food production for sustainability amid global population growth. They prioritize fostering food knowledge for health and environmental harmony [7]. Slow food, now a sociopolitical initiative, challenges corporate dominance and values small enterprises, cultural diversity, and biodiversity. It emphasizes social and environmental responsibility, supporting local employment and safeguarding traditional industries, small farms, food diversity, and the ecosystem [8]. This comprehensive approach encompasses economic, social, cultural, and environmental dimensions, with a focus on educating about agricultural practices and ecological preservation [9].

The slow food movement is intricately linked to SDG2, Zero Hunger. Slow food seeks to establish sustainable and resilient food systems that actively contribute to the realization of SDG2. The movement champions the principles of "good, clean, and fair" food, emphasizing quality, sustainability, and social justice in food production and consumption. Through initiatives that encourage the revival of local food and traditional cooking, support small-scale producers, and advocate for the conservation of cultural and biological diversity, slow food aligns itself with the objective of zero hunger while fostering local socio-environmental sustainability [10]. Agricultural extension is crucial for sustainable development and advancing the slow food movement's objectives [11–15]. It plays a pivotal role in propelling agriculture and disseminating technological progress beyond technical aspects, necessitating adept resource management [16]. With responsibilities in fostering growth, alleviating poverty, and addressing economic and social dimensions [17], agricultural extension is integral to establishing sustainable agriculture. However, practical models for agricultural extension in areas like wholesome nutrition are lacking. The current agricultural extension system faces challenges in policy formulation, organization, target clientele, and funding [14], hindering its capacity to fulfill various roles, including nutritional guidance. The lack of effective accountability results in predicaments diminished productivity, and reduced well-being for agricultural users and their households. Given Iran's rich cultural, climatic, and culinary diversity, coupled with the global expansion of cultural tourism, the nation has the potential to be a significant participant in the slow food movement [18]. Iran's expansive geographical expanse, diverse array of agricultural products, and culturally abundant tapestry directly influence local cuisine. Therefore, researching the constituents and elements of the slow food movement, while identifying efficacious extension methodologies, is imperative [19].

The development of a comprehensive model, considering economic, social, and cultural contexts, along with cultural sensitivities and the consequences of slow food consumption on the environment, culture, and health, is of paramount importance. Additionally, the absence of a well-defined developmental framework and the accompanying uncertainties regarding the components and elements of the slow food paradigm underscores the necessity for addressing these gaps through further investigation. Consequently, the primary research question guiding this study was the following: How can a comprehensive model be effectively designed for advancing slow food through the extension system, and which components and elements should this model encompass?

Slow food embodies dimensions promoting biodiversity, adopting a noble philosophy, championing ecologically mindful practices, preserving traditional methods, and advocating for wholesome eating. These dimensions are explored in many studies [20–27].

Numerous studies have explored the impacts of slow food development, covering aspects, such as disease prevention and ecological conservation [25,27,28]. Concurrently, conditions fostering slow food's evolution, including disorder, inadequate communication, steep costs, and a feeble public culture, have been identified [27–31]. Government policies, incentives, and heightened oversight are seen as pivotal in advancing slow food [21,32–34]. Strategies for cultivating slow food's progress have been discussed in studies by [22,32,35,36]. In Aşkin Uzel's [37] investigation, a conclusive determination was reached: slow food represents a valuable and sustainable approach to nourishment, holding the potential to alleviate illnesses stemming from suboptimal nutrition and enhance overall well-being. The propagation of slow food practices can also drive economic and societal progress in regions reliant on food production. To effectively advance slow food, education and the cultivation of wholesome eating habits, particularly among the younger demographic, are critically important. In the study by Bashiri et al. [38], the authors observed a significant reduction in the risk of cardiovascular diseases, diabetes, obesity, and digestive ailments through the adoption of slow food. This dietary approach further exerts favorable influences on cognitive health, diminishing anxiety and stress, augmenting concentration, and enhancing memory. Slow food reinforces the immune system, preempting related

maladies. Thus, as a salubrious and sustainable dietary paradigm, slow food occupies a pivotal role in the preservation and enhancement of human health.

Abbasi et al. [39] study explores the impact of slow food on sustainable development, emphasizing its relevance as the global population grows. Slow food is an environmentally conscious approach, reducing waste, managing resources efficiently, and as Shujaei et al. [40] noted promoting healthful food generation. It fosters economic and social progress in food-dependent regions, holding promise for societal advancement. Zerehposh et al. [41] found slow food significantly affects both physical and mental well-being, while Hafizi et al. [42] delved into its principles and methodologies, highlighting sustainability in production and consumption. Integrating slow food tenets, such as consuming natural and seasonal products and adopting health-conscious cooking techniques, promotes sustainable food practices. Ghorbani [43] demonstrated that implementing slow and low-fat food methodologies can improve health outcomes, reduce waste, stimulate local production, and generate employment. Facing challenges, Habibi et al. [44] emphasized the potential for slow and low-fat food principles to enhance community health and local progress. Masoudi et al. [45] underscored the positive influence of slow food on societal well-being, while Hafizi et al. [42] outlined challenges and proposed remedies, such as heightened awareness and education. Table 1 summarizes perspectives of previous scholars on effective factors in the slow food concept.

**Table 1.** Slow food concepts and key influencing factors.

| Key Influencing Factors | Authors |
|---|---|
| Slow food development strategies. | [20,22,29,36,37] |
| Government incentives, support policies, and the slow food movement. | [21,33–35] |
| The absence of planning, inadequate communication, high costs, and a deficient public culture serve as background conditions for the development of slow food. These encompass both background and causal conditions. | [25,27–31,35] |
| Consequences of slow food development include disease prevention and environmental protection. | [25,27–29,35] |
| Components supporting biodiversity, fostering a noble philosophy, embracing environmentally friendly food practices, reverting to ancestral traditions, and promoting healthful eating. | [20–22,24–27] |
| The utilization of local traditional foods in food and nutrition policies can stabilize food systems. However, the scarcity of authentic local cuisine in regions, coupled with the rise of foreign eateries, impacts the availability of healthful nutrition and contributes to the long-term risk of cardiovascular diseases. Irregular eating habits and excessive fluid consumption during meals are linked to increased odds of general and abdominal obesity. Dietary patterns, healthy eating profiles, and traditional cardiovascular disease risk factors are interconnected. Additionally, the consumption of high-calorie foods, such as fatty dairy products and red meats, highlights the importance of food tourism, nutritional habits, health education, and traditional approaches. The dynamic between traditional, sustainable food systems and contemporary dietary preferences is influenced by culture and convenience. | [18,46–52] |

## 2. Research Methodology

This research takes an applied, non-experimental approach, focusing on understanding the perspectives and insights of experts in slow food extensional prerequisites. Using a mixed methods design that combines qualitative and quantitative paradigms, this study aims to provide comprehensive results. To address the research questions, a descriptive survey strategy was employed. The qualitative sampling methodology in this research employs a comprehensive approach to ensure a nuanced understanding of slow food extensional prerequisites. The selection of participants involves a dual strategy, integrating both theoretical and purposive sampling techniques with a specific emphasis on the snowball method. Theoretical sampling enables the identification of individuals possessing theoretical knowledge and expertise in slow food extensional requirements, ensuring a well-rounded representation. Simultaneously, purposive sampling allows for the intentional selection of participants based on their relevance and significance to the research

objectives. The snowball method, a key component of this sampling strategy, facilitates the expansion of the participant pool. Starting with an initial set of participants, additional individuals are identified through referrals from the initial participants. This iterative process helps uncover hidden expertise and ensures a diverse range of perspectives within the domain of slow food extension. In the initial phase of the study, fifteen semi-structured interviews were carefully conducted. This number was determined based on the principle of theoretical saturation, indicating the point at which new information ceases to emerge, ensuring a thorough exploration of the subject matter. The semi-structured nature of the interviews allows for flexibility, enabling the exploration of unexpected insights and ensuring a holistic understanding of slow food extensional prerequisites. The results of this interview and the identified concepts for slow food extension requirements are presented in Table 2 and Figure 1. The results of this stage were considered for the quantitative phase.

Referring to the statistical population, it encompasses experts and managers extensively involved in agricultural extension and education. Affiliated with coordination management offices, agricultural research, and education centers nationwide, the sample size for this research consists of 218 actively engaged individuals in agricultural extension.

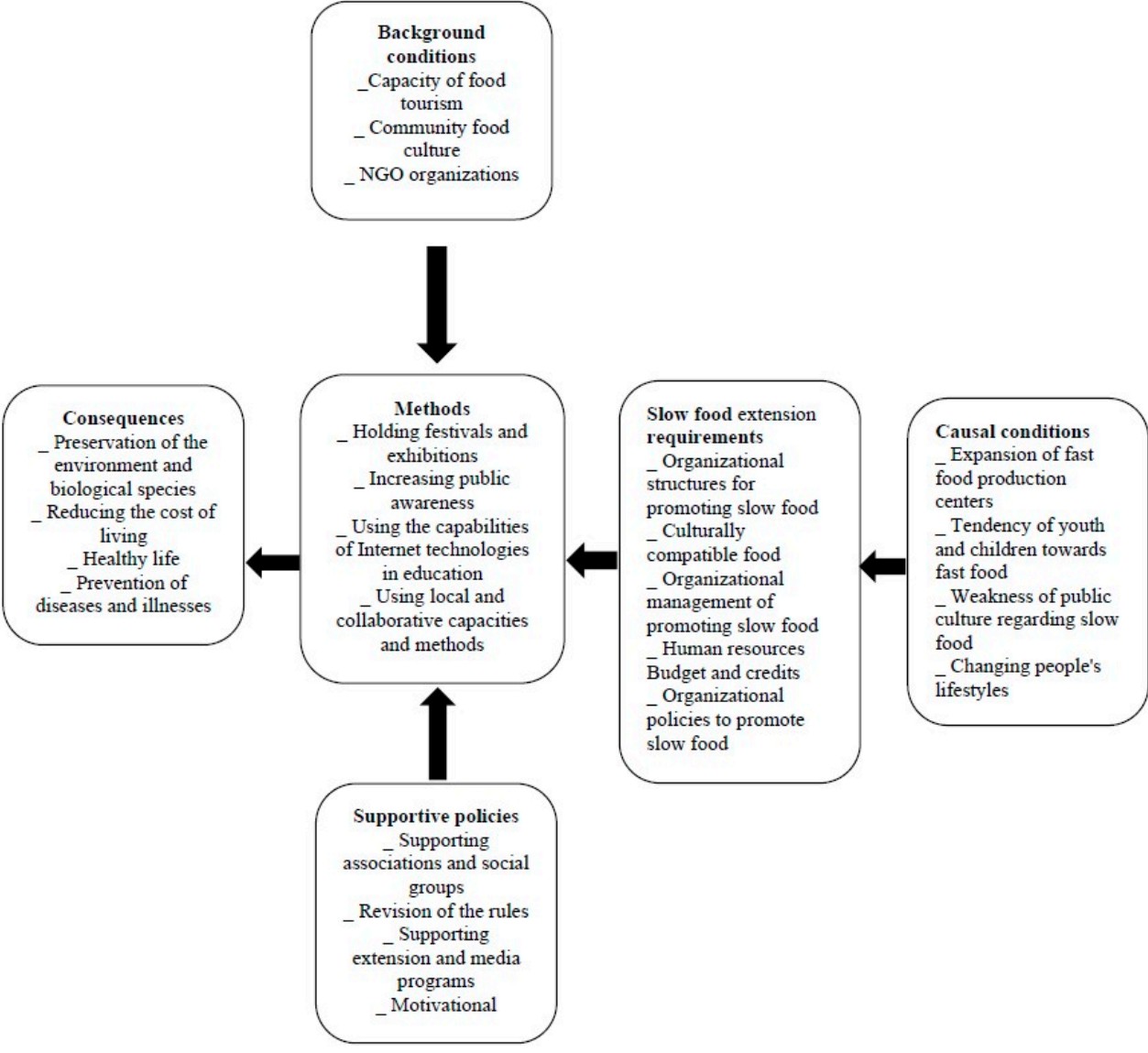

**Figure 1.** Main and major components of slow food extension.

**Table 2.** Identified concepts for slow food extension implications from semi-structured interviews.

| Identified Relevant Key Concept | Main Construct |
|---|---|
| Conformity with values and traditions, matching with morals, believing in the philosophy of good (delicious) food, promoting the taste of native and local foods, and financial and human structures. | Extension of slow food |
| Obtaining opinions from family members, obtaining opinions from relatives and friends, consulting slow food experts, obtaining opinions from teachers, and introducing and extension a suitable lifestyle. | Extension methods of slow food |
| Increasing government investment in education and extension of slow food and government support for popular and non-governmental organizations in the field of slow food. | Supportive policies (intervening conditions) |
| Existence of extension facilities in urban and rural environments, the existence of extension guidelines and instruction, and the existence of educational and extension centers and institutions. | Background conditions |
| Reducing the costs of treating diseases, reducing cardiovascular diseases, more communication between family members, and improving human health due to eating slow food. | Consequences of slow food extension |
| Air pollution in food production, use of polluted water in food production, and excessive use of chemical inputs in agriculture. | Challenges facing the use of slow food |

In the subsequent quantitative stage, descriptive research methodology was employed alongside a structural equation model. The initial use of factor analysis assists in discerning pivotal components and their corresponding significance coefficients. Subsequently, the structural equation model serves as a robust tool to unravel the intricate relationships among these components. Ultimately, this model culminates in presenting the definitive framework for slow food extensional prerequisites.

In this stage, the primary research tool was a researcher-made questionnaire consisting of two main sections based on the results of the qualitative phase. The initial section captured respondents' demographic characteristics, encompassing age, gender, level of education, field of education, organizational position, and work experience. The second section of the questionnaire focused on the promotional aspects of slow food. Comprising 75 items presented in a 5-level Likert scale format, this part aimed to gauge respondents' opinions on various dimensions. These dimensions encompassed Extension of Slow Food, Methods for Promoting Slow Food, Supportive and Motivational Policies, Background Conditions, Consequences of Slow Food, and the Causal Conditions of Slow Food.

To assess construct validity, we computed the average variance extracted index (AVE), indicating the extent to which the indicators contribute to the variance of the studied construct. AVE serves as a metric for construct validity, also recognized as convergent validity (Table 7). On the quantitative side, reliability was evaluated using both the Cronbach's alpha test and composite reliability for the items designed to measure the variables. It is important to note that the pre-test stage had a smaller sample size, making it inadequate for calculations within a structural equation model. Therefore, for assessing reliability during the pre-test stage, we employed Cronbach's alpha (Table 3). In contrast, during the model test stage, we adopted the CR method. Notably, the main questionnaires provided a more substantial sample size, enabling the determination of reliability through the CR approach.

**Table 3.** Cronbach's alpha coefficient for questionnaire dimension.

| Row | Questionnaire Dimensions | Number of Items | Cronbach's Alpha Coefficients |
|---|---|---|---|
| 1 | Extension of slow food | 14 | 0.781 |
| 2 | Methods of promoting slow food | 21 | 0.698 |
| 3 | Supportive policies | 10 | 0.895 |
| 4 | Background conditions | 10 | 0.887 |
| 5 | Consequences of slow food | 8 | 0.741 |
| 6 | Causal conditions of slow food | 12 | 0.952 |

The general fit index was introduced to check the fit of the model. The overall criterion of fit (GoF) (Equation (1)) can be obtained by calculating the geometric mean of the shared mean and $R^2$.

$$\text{GoF} = \sqrt{\text{average (Commonalities)} \times R^2} \qquad (1)$$

In order to assess the suitability of the data for factor analysis, it is necessary to conduct a test of sampling adequacy. This test is measured using the Kaiser-Meyer-Olkin (KMO) indicator (Table 4).

**Table 4.** Sample size adequacy test.

| KMO and Bartlett's Test | | |
|---|---|---|
| Kaiser-Meyer-Olkin Measure of Sampling Adequacy. | | 0.876 |
| Bartlett's Test of Sphericity | Approx. chi-square | 6151.33 |
| | d.f. | 217 |
| | *p*-value | 0 |

## 3. Results

The analysis of the subjects' age revealed the highest frequency in the "41–50 years" age group, comprising 86 individuals (39.5%) with an average age of 42.69. The majority of participants were male, accounting for 172 individuals (78.9%). Examining the subjects' education level, the most common category was "master's degree" with 100 individuals (45.8%), while the field of "agricultural engineering" had the highest frequency at 119 individuals (54.6%). Regarding agricultural experience, the most frequent range was "11–15 years" with 144 individuals (36.6%), and the average experience was 12.99 years. In terms of organizational positions, the highest frequency was observed among "experts" with 109 individuals (50%), whereas "academic faculty" had the lowest frequency of 24 individuals (11%) (Table 5).

**Table 5.** Respondents demographic characteristics (*n* = 218).

| Feature | Group | Frequency | Percentage |
|---|---|---|---|
| Age (years) M = 40.46 | 20–30 | 41 | 18.8 |
| | 30–40 | 55 | 25.2 |
| | 40–50 | 86 | 39.5 |
| | 50–60 | 36 | 16.5 |
| Gender | Mail | 172 | 78.9 |
| | Female | 46 | 21.1 |
| Level of education | B.Sc. | 30 | 13.8 |
| | M.Sc. | 100 | 45.8 |
| | Ph.D. | 88 | 40.4 |
| Field of study | Technical engineering | 20 | 9.2 |
| | Basic sciences | 22 | 10.1 |
| | Agricultural engineering | 119 | 5436 |
| | Humanities | 41 | 18.8 |
| | Other | 16 | 7.3 |
| Organizational position | Expert | 109 | 50 |
| | Senior expert | 45 | 20.7 |
| | Manager | 40 | 18.3 |
| | Faculty members | 24 | 11 |
| Work experience (years) | 1–5 | 26 | 11.9 |
| | 5–10 | 63 | 28.9 |
| | 10–15 | 73 | 33.5 |
| | 15–20 | 44 | 20.2 |
| | 20–25 | 7 | 3.2 |
| | 25–30 | 5 | 2.3 |

## 4. Measurement Model Test

The model testing process is conducted step by step, beginning with the construction of the PLS model in the software. This step helps to identify the relationships between variables and their corresponding indicators and constructs. In the initial phase of our research, we employed the grandad theory paradigm model. Through face-to-face interviews with key influencers, we extracted essential elements, including causal conditions, background conditions, intervening conditions, strategies, and consequences associated with the extension of slow food. It is crucial to highlight that these identified factors directly align with the variables we are investigating in our study (refer to Figure 1). Based on the known variables in Table 2 and the relationship between them according to the grounded theory model (Figure 1), the model was evaluated. As illustrated in Table 6 and Figure 2, the causal conditions variable demonstrates the most significant causal effect when estimating the variance of slow food (beta = 0.667, t = 3.237, $p < 0.05$). For this factor, people's lifestyle, the tendency of young people and children to eat fast food, and fast food production centers had the highest relationship with casual conditions of slow food. Subsequently, supportive policies exhibit the highest meaningful causal relationship with slow food in next step (beta = 0.349, t = 2.671, $p < 0.05$).

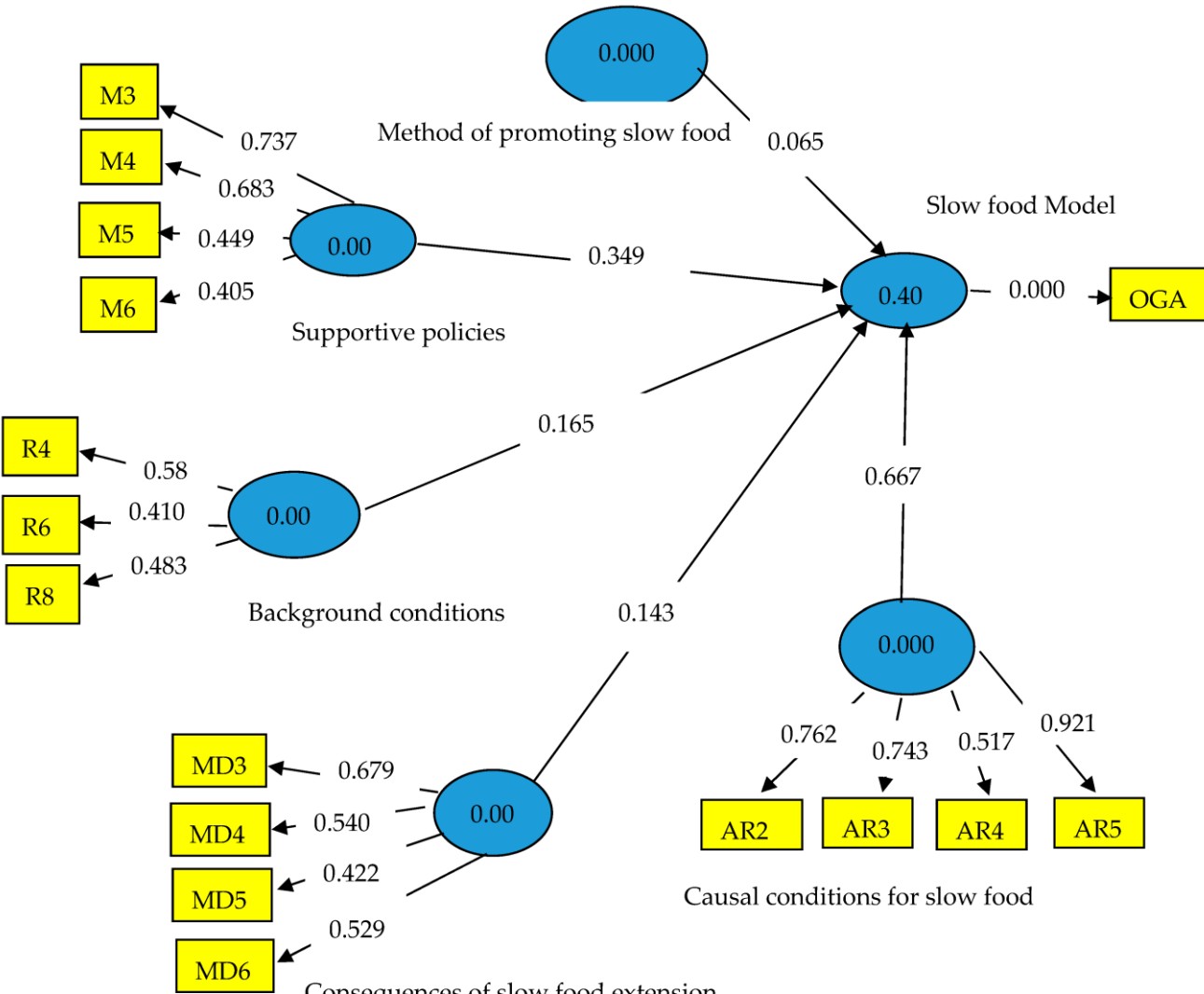

**Figure 2.** Measurement of the final model and the results of the hypotheses in the standard mode.

**Table 6.** Values of factor loading under the modified components of the knowledge and information system of smallholder farms.

| Factors | Visible Variables | Symbols | Factor Loading | t-Value |
|---|---|---|---|---|
| Background conditions | Capacity of food tourism | R4 | 0.584 | 2.618 * |
| | Community food culture | R8 | 0.483 | 4.319 * |
| | Slow food NGOs | R6 | 0.410 | 3.348 * |
| Causal conditions | People's lifestyle | AR5 | 0.921 | 3.704 * |
| | The tendency of young people and children to fast food | AR2 | 0.762 | 2.381 * |
| | Fast food production centers | AR3 | 0.743 | 2.127 * |
| | Weakness of general culture | AR4 | 0.517 | 2.227 * |
| Intervening conditions | Revision of the rules | M3 | 0.737 | 3.349 * |
| | Supporting extension and media programs | M4 | 0.683 | 3.177 * |
| | Supporting associations and social groups | M5 | 0.449 | 3.012 * |
| | The support of relevant groups | M6 | 0.405 | 2.420 * |
| Extension variables | Holding festivals and exhibitions | A9 | 0.812 | 13.102 ** |
| | Using the capacity of non-governmental organizations | A10 | 0.765 | 10.106 ** |
| | Use of mass media | A13 | 0.723 | 10.890 ** |
| | Using local and popular capacities and methods | A10 | 0.651 | 10.106 * |
| | Using the capabilities of Internet technologies in education | A11 | 0.495 | 4.880 * |
| Consequence's variables | Prevention of diseases and illnesses | MD3 | 0.679 | 3.361 * |
| | Healthy life | MD4 | 0.540 | 2.544 * |
| | Reducing the cost of living | MD6 | 0.529 | 4.059 * |
| | Environmental protection | MD5 | 0.422 | 2.097 * |

** $p < 0.01$, * $p < 0.05$.

## 5. Factor Load Measurement

The reliability of each item is ascertained based on the count of factor loadings linked to each observed variable. This evaluation aids in determining the efficacy of the observed variables in gauging the latent, underlying variables. Typically, a minimum acceptable value of 0.3 is deemed suitable. Factor loadings of 0.4 signify a moderate level of significance, and those exceeding 0.5 denote a substantial level of significance. Table 7 presents the factor loading values corresponding to each of the independent variables.

The high fit of the model indicates that the model is well explained. Esposito et al. [46] suggest that a goodness-of-fit index (GoF) higher than 0.5 indicates a good fit for the model. Davari and Rezazadeh [47] agree with this assessment. The overall GoF for this research model is 0.503, which suggests the model has a good fit (Table 8).

**Table 7.** General model quality criteria.

| Components | Composite Reliability (CR) | Coefficient of Determination (R²) | Composite Reliability | Cronbach's Alpha | Common Values (Community) | Shared Reliability (AVE) | Redundancy Index Q² ( = 1 − SSE/SSO) |
|---|---|---|---|---|---|---|---|
| Extension of food slow | 0.817 | 0.389 | 0.802 | 1.00 | 1.000 | 1.00 | 0.298 |
| Method food slow | 0.895 | 0.573 | 0.931 | 0.768 | 0.742 | 0.823 | 0.447 |
| Supportive policy food slow | 0.912 | 0.573 | 0.903 | 0.839 | 0.854 | 0.805 | 0.176 |
| Background food slow | 0.954 | 0.573 | 0.851 | 0.920 | 0.951 | 0.782 | 0.341 |
| consequences food slow | 0.861 | 0.573 | 0.911 | 0.832 | 0.789 | 0.841 | 0.177 |
| Causal conditions of slow food | 0.924 | 0.108 | 0.912 | 0.736 | 0.874 | 0.766 | 0.188 |

**Table 8.** Final model fit.

| Index | R² | Communality |
|---|---|---|
| Extension of slow food | 0.62 | 0.56 |
| Extension methods | 0.56 | 0.37 |
| Supportive policies | 0.75 | 0.43 |
| Background conditions | 0.88 | 0.16 |
| Benefits of slow food | 0.56 | 0.38 |
| Causal conditions of slow food | - | 1 |

## 6. Discussion

The study unveils six crucial elements within the structure of the slow food extension framework. These elements encompass requirements, extension, background conditions, supporting policies, consequences, and causal conditions components. Employing the research background and grounded theory approach, we meticulously crafted a distinctive conceptual model, making a noteworthy contribution to the field. This conceptual model underwent empirical testing within a statistical population comprising experts and extension specialists nationwide. The analysis results affirmed its appropriateness and effectiveness as a suitable framework.

The study indicates that advancing slow food necessitates enhancing human resources, a fundamental aspect of this initiative. Extending slow food further requires dedicated budgetary and financial resources, constituting essential sub-components. This aligns with Allahyari's findings [14], emphasizing the substantial investment required for slow food extension. Efficient organizational structures are crucial in the array of slow food model prerequisites, as supported by various studies [26,53,54]. Allahyari's research [14] reinforces the idea that slow food extension mandates a significant infusion of financial resources and adept budgeting. Collectively, this evidence emphasizes the pivotal role these sub-components play within the slow food model, working in tandem with human resources.

The research supports the use of virtual networks, aligning with Fatemi Amin and Fouladian's findings [53]. Both previous researchers and experts interviewed highlight the importance of people's awareness and knowledge. Slow food aims to foster active engagement, emphasizing that informed consumers understand food production challenges [55]. Building on background research, the study's findings on extensional methods align with MirKarimi et al. [54] and Williams et al. [56]. Other studies by Aşkin Uzel [37], Petrini [35],

Slow Food [20], Simonetti [22], and Heitmann et al. [36], also discuss methods for slow food development.

Experts in agricultural policies emphasize endorsing organic agriculture and reducing pesticide use within the slow food movement. Policies on transgenic crops or seed modification, a key concern addressed in this research, are also deemed essential. Peano et al.'s study [55] suggests that establishing slow food committees enhances sustainability, focusing on socioeconomic and cultural aspects while prioritizing environmental and qualitative aspects of food production. Related research by Leitch [33], Slow Food [20], Dumitru et al. [21], and Schneider [34] aligns with this study's findings, emphasizing the importance of advocating for government policies and expanding oversight for the advancement of slow food [20].

The quantitative phase of the research, utilizing partial least squares analysis, affirmed the influence of tourism, traditional food culture, and non-governmental organizations in expanding the slow food model. These results align with various studies, including those by Slow Food [20], Dumitru et al. [21], Simonetti [22], Counihan and Van Esterik [24], Andrews [25], Petrini [26], and Sassatelli and Davolio [27]. Durst and Bayasgalanbat's [57] research further supports these findings, while Abshar [47] and Aini Zeinab and Sobhani [48] demonstrate that traditional and indigenous foods in Iran are more environmentally sustainable than popular Western foods.

The background research aligns with Taghvi [58] and Peano et al. [55], reinforcing and supporting the discussed issue. The prevalence of fast food in societies is partly attributed to employed family members with limited time for food preparation due to work or studies.

The growing interest in healthy eating acknowledges slow food as a health-conscious pattern that prioritizes natural ingredients while avoiding preservatives. This approach is bolstered by strategies, such as promoting organic agriculture and endorsing a diverse diet, which is gaining traction for its health and eco-friendly attributes. By highlighting the use of fresh, raw, and natural ingredients, slow food contributes to preventing heart diseases, diabetes, and cancer; reducing food waste; improving product quality; and fostering sustainable agriculture. The promotion of supportive policies, including financial incentives, stimulating consumer demand, and establishing suitable retail spaces, is essential for its widespread adoption.

To establish a foundation for slow food principles, key elements must be addressed: cultivating technical proficiency among farmers, establishing a favorable market, and enhancing the national food system. Three pivotal factors—food tourism capacity, community food culture, and Non-Governmental Organizations (NGOs) involvement—are suggested. Food tourism capacity involves a region's ability to offer local culinary delights, preserving traditions, promoting sustainable practices, and cultivating local craftsmanship. Food culture, tied to nutrition habits and behaviors, endorses slow food by promoting local and organic fare, encouraging home cooking with fresh ingredients, celebrating food diversity, and enhancing the social experience of consuming food. Leveraging existing food culture fosters a deeper understanding, guiding individuals toward healthier choices and embracing the slow food philosophy.

NGOs play a vital role in promoting local and slow food principles among farmers and producers through activities like training sessions, awareness campaigns, and workshops, utilizing internet resources and collaborative approaches. They also organize classes and workshops to encourage individuals to adopt home-cooked meals, promoting diverse and high-quality foods. Beyond education, these organizations conduct research to raise awareness and advocate for slow and healthy food principles, contributing to the widespread adoption of sustainable eating habits and the preservation of local culinary traditions.

Slow food practices yield several key benefits, including the prevention of nutrition-related diseases, cost reduction in maintaining a healthy lifestyle, and contributions to environmental conservation. These practices effectively combat cardiovascular diseases, diabetes, cancer, and obesity by promoting mindful eating and reducing risks associated with excessive calorie and fat consumption. Embracing slow food translates to healthier

living, diminishing the financial burden of disease prevention and treatment, streamlining food-related expenses, and enhancing overall quality of life. Ecologically, the implications are significant; local, fresh ingredients minimize long-haul food transportation, averting environmental harm from preserved and processed goods and playing a crucial role in preserving biodiversity and preventing environmental degradation.

To expand slow food practices, a well-suited organizational framework is crucial. This framework should include culturally aligned nourishment, adept management of slow food extension, adequate human resources, budget allocations, and organizational policies championing healthful cuisine. Key factors for success include robust leadership committed to slow food promotion, a dedicated and proficient team, provision of necessary resources, and collaborative engagement with public and private entities, research centers, and universities.

Cultural compatibility of food is essential for slow food extension, aligning with local tastes and safeguarding indigenous knowledge, values, and traditions. This requires considering prevalent culture, historical context, and local culinary traditions. Culturally compatible foods serve as communication and social catalysts within society, nurturing and reinforcing social bonds.

Efficient management of slow food promotion involves forming an expert working group to devise strategies aligned with audience needs. Educational initiatives, workshops, festivals, and collaboration with organizations are pivotal. Human resources encompass education, awareness, and research, with encouragement and support being crucial. Financial allocation and suitable facilities are pivotal considerations for successful slow food promotion, contributing to the overall well-being of society.

Supportive policies are integral to societal benefit, particularly in the context of slow food promotion. A key policy avenue involves revising laws related to food, nutrition, and health, offering a streamlined means to enhance the reach of the slow food movement. Amendments and updates to these laws facilitate the process of promoting slow food effectively. Extensional programs, encompassing various media outlets, such as the internet, television, and radio, play direct and indirect roles in this promotion. Financial backing and well-organized execution of these programs significantly contribute to the success of the slow food movement. Additionally, associations and social groups contribute by hosting festivals, exhibitions, workshops, and educational webinars focused on slow food.

Causal conditions and factors drive the adoption of slow food, including the proliferation of fast food production centers, the inclination of young individuals and children toward fast food, limited emphasis on slow food in public culture, and shifts in people's lifestyles. Given the health risks associated with fast food, curtailing fast food production centers and promoting slow food becomes crucial for public health. Increasing awareness, fostering slow food production, and enacting laws pertaining to fast food production and slow food promotion are necessary. Addressing the deficiency in public culture involves boosting understanding of the benefits of slow food for overall health. Strategies for extending slow food adoption include training initiatives, incentives for slow food production, menu adjustments, and innovative advertising techniques.

Globally, the slow food movement is recognized as an international force in nutrition and agriculture. Actively supporting sustainable agriculture, high-quality food markets, dietary diversity, and healthy eating habits, this movement contributes to better health, improved quality of life, reduced food waste, environmental preservation, and economic opportunities. Raising awareness and advocating for slow food assumes great importance on a global scale, addressing vital concerns in the domains of nutrition and health for the worldwide community.

## 7. Conclusions

The extension of the slow food movement holds paramount importance as a developmental imperative in Iran. To bolster its reach, a comprehensive examination of factors is essential, encompassing compatible extension requisites, methods for proliferation, sup-

portive policies, underlying and causal conditions, as well as the ensuing consequences of adopting slow food practices. Given the profound influence of slow food on promoting health-conscious eating habits and enhancing people's overall quality of life, a concerted push to amplify its adoption on a national scale becomes imperative.

The positive consequences of endorsing slow food practices, benefiting societal health and fueling economic growth, underline the need for heightened efforts on a national scale. A collaborative effort among stakeholders—producers, consumers, government bodies, and private institutions—is essential to advance the slow food cause.

To optimize the role of extension institutions, empowering human resources within the Agricultural Extension and Education Institute and the Ministry of Health's Food and Drug Deputy is crucial. This entails formal endorsement, resource allocation, and elevating knowledge in the realm of healthy and slow food.

Media integration is recommended by positioning the "healthy life" theme within the scope of "healthy food" through dedicated programs on mass media platforms. Enhanced monitoring of fast food establishments is warranted to ensure adherence to standards.

Supporting non-governmental entities dedicated to slow food, coupled with strategic planning, resource allocation, and organizational fortification within government institutions, is crucial. Organizing food festivals, collaborating between extension and health systems, and emphasizing organic practices align with agricultural policies.

Embracing these recommendations reinforces the slow food movement, contributing to healthier lifestyles, sustainable agriculture, and improved societal well-being.

## 8. Recommendations

To bolster the slow food movement and enhance the role of extension institutions, the following recommendations are proposed:

*Empowering Extension Institutions:*

- Given the pivotal role of the Agricultural Extension and Education Institute and the Ministry of Health's Food and Drug Deputy in the slow food extension framework, formal endorsement and meticulous planning are crucial.
- Resources should be allocated to enhance human resources' knowledge in healthy and slow food practices, positioning human resource empowerment as a linchpin in the broader domain of human resources management.

*Media Integration:*

- Positioning the "healthy life" theme within the scope of "healthy food" on mass media platforms, including radio and television, is recommended.
- A designated program promoting healthy food and lifestyles should be crafted, approved by the Supervisory Council of the Broadcasting Organization, and prominently featured in national-level strategic documents.

*Enhanced Monitoring:*

- Rigorous and continuous supervision of fast food production and distribution establishments is necessary to ensure adherence to standards. A comprehensive plan supporting this effort is warranted.

*NGO Support:*

- Government institutions should actively encourage and endorse the establishment of non-governmental entities dedicated to slow food, supporting extension efforts through careful planning, resource allocation, and organizational fortification.

*Healthy Eating Promotion:*

- Organizing food festivals through collaboration between extension and health systems, coupled with robust education on healthy nutrition, effectively nurtures wholesome eating habits.
- Aligning the slow food movement with agricultural policies emphasizing organic practices is crucial for promoting sustainable agricultural practices.

Embracing these recommendations will reinforce the slow food movement, contributing to healthier lifestyles, sustainable agriculture, and improved societal well-being.

**Author Contributions:** Methodology, M.S.S.; Validation, M.N.; Investigation, H.N., M.S.A. and E.D.; Writing—original draft, M.S.A.; Supervision, M.S.S. and M.S.A. All authors have read and agreed to the published version of the manuscript.

**Funding:** This research received no external funding.

**Institutional Review Board Statement:** Not applicable.

**Informed Consent Statement:** Written informed consent has been obtained from the patient(s) to publish this paper.

**Data Availability Statement:** The data presented in this study are available on request from the corresponding author.

**Conflicts of Interest:** The authors declare no conflict of interest.

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
