# Peer review of "Exploring Extension Implications for Slow Food Development in Iran: A Comprehensive Analysis"

_sustainability, doi:10.3390/su152316538_

Round 1
Reviewer 1 Report
Comments and Suggestions for Authors
The paper focuses on the implications for Slow Food Development in Iran. However, there are a few comments that should be addressed. The comments are as follows:
1- The overall objective of the work is not clear.
2 - In the introduction section, the authors should spend more ink to better explain their added value and contribution of the paper to the academic literature. Furthermore, try to provide a more insightful analysis of previous studies and some of their shortcomings.
3 - What are the empirical assumptions/ hypotheses that back your research questions? This needs to be addressed carefully.
4 - You need to describe all variables included in the analysis and the rationale for this in line with the literature.
5 - The authors should improve the discussion of results in light of previous findings as well as try to provide the economic intuition of results.
6 - Comments on the economic significance of the results and state if your results are consistent with the literature and theory background or not.
7 - The authors should elaborate on the main storyline and the policy implications. Furthermore, provide some limitations on your analysis and how they open the doors for future research.
8 – The authors should discuss the policy implications for your results. Please make them cristal clear.
9 - Please carefully proofread the manuscript.
Comments on the Quality of English Language
Please carefully proofread the manuscript.
Author Response
1- Added line 75-84
2 - Added line 50
3 - We eliminated the sub-hypothesis. As stated earlier, we conducted a qualitative study, and subsequently, we examined its findings quantitatively using SEM.
4 - Added lines 150-153 and Table 2 and Fig 4.
5 - Rewrittenn.
6 - This study aims to find a model that is not based on a theory.
7 - Improved
8 – improved
9 - done
Please see detailed revisions highlighted in yellow in the attached manuscript file.

Reviewer 2 Report
Comments and Suggestions for Authors
The manuscript is interesting and the authors address a topical issue.
The manuscript presents many new points, but I think it should be improved by:
1. rewriting the introduction and introducing a literature review chapter.
2. the clear formulation of the hypotheses
3. the introduction of references to tables in the text
4. explanation of the data in all the tables
5. fig 1 and fig 2 must be clearer and require explanations
6 pay attention to the numbering of the figures, figure 3 does not exist in the manuscript
7. make references to figures
8. Figure 4 on which information/data did you build it
9. present the limitations of the study
10. create the bibliography in accordance with the rules of the journal
Author Response
1. done
2. This study is not theoretical. Instead, it aims to develop a model without a hypothesis. We simply test the results in the qualitative phase of this process.
3. Done
4. Done
5. Fig2 is deleted and is data is added to Table 6
6 Done
7. Done
8. Done
Please see detailed revisions highlighted in yellow in the attached manuscript file.

Reviewer 3 Report
Comments and Suggestions for Authors
the paper titled; Exploring Extension Implications for Slow Food Development 2 in Iran: A Comprehensive Analysis, is very important and it has added value to the filed. However, there are some important points to be improved :
The introduction is very long, it is better to reduce this section and fit it with the objective of the study.
The objective is not clear and must be simple for the reader
the methodology is not clear, more details are needed
the results mus be developed
the discussion must be improved and
the reference in the text must be a number and not the author name ( see guide to authors)
Author Response
The objective is not clear and must be simple for the reader
the methodology is not clear, more details are needed: Completed in lines 132-153 and 167-176
the results must be developed: Done in the yellow part in the result section
References are arranged based on journal format.
Please see detailed revisions highlighted in yellow in the attached manuscript file.

Reviewer 4 Report
Comments and Suggestions for Authors
My comments related to improving the article are as follows:
Point 1: Why are the main aspects of slow food mode these 6 aspects and how are they derived?
Point 2: Is the candidate participating in the survey representative? Is the concept of slow food expansion requirements reasonable in semi-structured interviews, and what does the M1 in Pundits represent in Table 3?
Point 3: The discussion section of the article is too verbose and needs to be concise.
Point 4: The format of the table in the text is not standardized, and a three-line table should be used.
Point 5: The image quality in the text is not high and cannot be seen clearly.
Point 6: The reference format is inconsistent, such as reference 3, 7, 8, 16 and so on. Please check on your reference format carefully and modify it.
Author Response
1- First, we conducted a qualitative study using a grounded theory approach, which involved semi-structured interviews. The insights gained from this phase were then utilized in the subsequent quantitative phase. line 167
Point 2: added in lines 132-157.
Point 3: done
Point 4: improved
Point 5: improved
Point 6: improved based on journal format.
Please see detailed revisions highlighted in yellow in the attached manuscript file.

Round 2
Reviewer 2 Report
Comments and Suggestions for Authors
The authors should consider a comprehensive revision of the entire paper, as the current revisions appear to be only partial and insufficient.
Figure 2 is absent, and the author is advised to review and address any relevant conditions associated with it.
Figure 3 is far too far from where it is called in the text L210. In addition, it is not explained.
The authors did not present the limitations of the study, as we initially recommended.
Furthermore, the discussion section of the article is overly narrative and would benefit from being more concise and to the point.
Lastly, the reference format displays inconsistencies.
Author Response
Figure 2 is absent, and the author is advised to review and address any relevant conditions associated with it. Figure 2 is removed from the text and its related data is added to Table 6 as a t-value column.
Figure 3 is far too far from where it is called in the text L210. In addition, it is not explained. Figure 3 is renamed to Figure 1 and moved to line 225. Also, it is explained in lines 211-216 (highlighted in green color in the text).
Furthermore, the discussion section of the article is overly narrative and would benefit from being more concise and to the point.
Please see the attachment for detailed revisions.

Reviewer 3 Report
Comments and Suggestions for Authors
The paper is well revised and can be suitable to be published and accepted
Author Response
Thanks for your considerations

Reviewer 4 Report
Comments and Suggestions for Authors
The author needs to revise the whole paper. My suggestions are as follows:
Comments#1 The author has only made partial revisions, which is not enough.
Comments#2 Figure 2 is missing and the author needs to check these conditions.
Comments#3 The discussion section of the article is too verbose and needs to be concise.
Comments#4 The reference format is inconsistent. Please check on your reference format carefully and modify it.
Author Response
Comments#2 Figure 2 is missing and the author needs to check these conditions. Figure 2 is removed from the text and its related data is added to Table 6 as a t-value column.
Comments#3 The discussion section of the article is too verbose and needs to be concise. We have extensively revised the discussion section, reducing its length from 3459 words to 1930 words.

Round 3
Reviewer 4 Report
Comments and Suggestions for Authors
The author adds all amendments, I have no further comments.